# Gene Loss and Evolution of the Plastome

**DOI:** 10.3390/genes11101133

**Published:** 2020-09-25

**Authors:** Tapan Kumar Mohanta, Awdhesh Kumar Mishra, Adil Khan, Abeer Hashem, Elsayed Fathi Abd_Allah, Ahmed Al-Harrasi

**Affiliations:** 1Biotech and Omics Laboratory, Natural and Medical Sciences Research Centre, University of Nizwa, Nizwa 616, Oman; adilsafi122333@gmail.com; 2Department of Biotechnology, Yeungnam University, Gyeongsan 38541, Korea; awadhesh.biotech07@gmail.com; 3Botany and Microbiology Department, College of Science, King Saud University, Riyadh 11451, Saudi Arabia; habeer@ksu.edu.sa; 4Mycology and Plant Disease Survey Department, Plant Pathology Research Institute, Giza 12511, Egypt; 5Plant Production Department, College of Food and Agricultural Sciences, King Saud University, P.O. Box. 2460, Riyadh 11451, Saudi Arabia; eabdallah@ksu.edu.sa; 6Natural Product Laboratory, Natural and Medical Sciences Research Centre, University of Nizwa, Nizwa 616, Oman

**Keywords:** chloroplast genome, plastome, evolution, deletion, duplication, recombination, nucleotide substitution

## Abstract

Chloroplasts are unique organelles within the plant cells and are responsible for sustaining life forms on the earth due to their ability to conduct photosynthesis. Multiple functional genes within the chloroplast are responsible for a variety of metabolic processes that occur in the chloroplast. Considering its fundamental role in sustaining life on the earth, it is important to identify the level of diversity present in the chloroplast genome, what genes and genomic content have been lost, what genes have been transferred to the nuclear genome, duplication events, and the overall origin and evolution of the chloroplast genome. Our analysis of 2511 chloroplast genomes indicated that the genome size and number of coding DNA sequences (CDS) in the chloroplasts genome of algae are higher relative to other lineages. Approximately 10.31% of the examined species have lost the inverted repeats (IR) in the chloroplast genome that span across all the lineages. Genome-wide analyses revealed the loss of the *Rbcl* gene in parasitic and heterotrophic plants occurred approximately 56 Ma ago. *PsaM, Psb30, ChlB, ChlL, ChlN,* and *Rpl21* were found to be characteristic signature genes of the chloroplast genome of algae, bryophytes, pteridophytes, and gymnosperms; however, none of these genes were found in the angiosperm or magnoliid lineage which appeared to have lost them approximately 203–156 Ma ago. A variety of chloroplast-encoded genes were lost across different species lineages throughout the evolutionary process. The *Rpl20* gene, however, was found to be the most stable and intact gene in the chloroplast genome and was not lost in any of the analyzed species, suggesting that it is a signature gene of the plastome. Our evolutionary analysis indicated that chloroplast genomes evolved from multiple common ancestors ~1293 Ma ago and have undergone vivid recombination events across different taxonomic lineages.

## 1. Introduction

Photosynthesis is a process by which autotrophic plants utilize chlorophyll to transform solar energy into chemical energy [1]. Almost all life forms depend directly or indirectly on this chemical energy as a source of energy to sustain growth, development, and reproduction of their species [2,3]. This essential process occurs inside a semiautonomous organelle, commonly known as a plastid or chloroplast [4]. Current knowledge indicates that the origin and evolution of plastids occurred through the endosymbiosis of ancestral cyanobacteria with nonphotosynthesizing cells that dates back to 1.5 to 1.6 billion years ago [5,6]. The subsequent divergence of a green plastid lineage occurred prior to 1.2 billion years ago and led to the development of land plants approximately 432 to 476 million years ago, and to seed plants around 355 to 370 million years ago [6]. A subsequent split into gymnosperms and angiosperms occurred approximately 290 to 320 million years ago and the divergence of monocots and eudicots within the angiosperm lineage occurred approximately 90 to 130 million years ago [6]. Throughout this evolutionary time scale, the endosymbiont retained its existence inside the cell and its dominant function of photosynthesis without undergoing any basic evolutionary changes (photosynthesis) [7,8,9,10]. In addition to photosynthesis, this semiautonomous organelle also plays an important role in the biosynthesis of amino acids, lipids, carotenoids, and other important biomolecules [11,12,13,14,15]. Studies indicate that the plastid genome has retained a complete set of protein-synthesizing machinery and encodes approximately 100 proteins [16]. All other proteins required by the chloroplast, however, are encoded by the nuclear genome. All of the protein synthesis and photosynthetic machinery used by the plastid is encoded by its own genome, commonly referred to as the plastome, that is arranged in a quadripartite structure [17,18,19,20]. The size of the plastid genome of land plants is reported to range from 120 to 190 kb [21,22,23]. The quadripartite structure consists of four main segments, referred to as the small single-copy region (SSC), large single-copy region (LSC), and the inverted repeat A and B (IR_A_ and IR_B_) regions [24]. The size of the IR region ranges from 10 to 15 kb in nonseed plants to 20–30 kb in angiosperms [24,25,26,27]. The IR_A_ and IR_B_ regions are reported to share a conserved molecular evolutionary pattern [28,29]. Studies also indicate that the genes in the plastome genome are organized in an operon or operon-like structure that undergoes transcription, producing polycistronic precursors [30]. The majority of genes in the chloroplast genome have been either functionally transferred to the nuclear genome or lost during evolution [31,32]. For example, the functional genes *tufA, ftsH, odpB,* and *Rpl5* have been transferred from the plastome to the nucleus [33,34]. Structural rearrangements of the plastid genome have occurred throughout its evolution; resulting in expansion, contraction, or loss of genetic content [23]. These events have occurred multiple times during the evolution of the chloroplast and can be specific to a single species, or sometimes to a whole plant order [25,35,36,37,38]. Changes in the architecture of the IR regions can affect the entire plastid chromosome and its immediate neighborhood. For example, several genes associated with the SSC region got duplicated, including *Ycf2,* due to the relocation of the IR region [23]. Although several analyses of the plastid genome have been conducted, a comprehensive comparative study of the plastid genome at a large-scale has not yet been reported. Comparative studies have thus far only included a few species of an order or a few species from a few different groups. Therefore, a large-scale analysis of 2511 chloroplast genomes was conducted to better understand the genomics and evolution of the plastid genome. Details of the novel genomic features of the chloroplast genome are reported in the present study. 

## 2. Materials and Methods

### 2.1. Sequence Retrieval and Annotation

All of the sequenced chloroplast genomes available up until December 2018 were downloaded from the National Center for Biotechnology Information (NCBI) and used in the current study to analyze the genomic details of the chloroplast genome. In total, 2511 full-length complete chloroplast genome sequences were downloaded, including those from algae, bryophytes, pteridophytes, gymnosperms, monocots, dicots, magnoliids, and protist/protozoa (Appendix A). All of the individual genomes were subjected to OGDRAW to check for the presence and absence of inverted repeats in the genome [39]. Genomes that were found to lack inverted repeats (IR), as determined by OGDRAW, were further searched in the NCBI database to cross verify the absence of IR in their genome. The annotated coding DNA sequences (CDS) sequences in each chloroplast genome were downloaded and the presence or absence of CDS from all chloroplast genomes were searched in each individual genome using Linux programming. Species that were identified as lacking a gene in their chloroplast genome were noted and further rechecked manually in the NCBI database. Each chloroplast genome was newly annotated using the GeSeq-annotation of the organellar genomes pipeline to further extend the study of gene loss in chloroplast genomes [40]. The combined analysis of NCBI and GeSeq-annotation of the organellar genomes were considered in determining the absence of a particular gene in a chloroplast genome. 

The CDS of the nuclear genome of 145 plant species were downloaded from the NCBI database. The presence of chloroplast-encoded genes in the nuclear genome was determined using Linux-based commands and collected in a separate file. The chloroplast-encoded genes present in the nuclear genomes were further processed in a Microsoft Excel spreadsheet. 

### 2.2. Multiple Sequence Alignment and Creation of Phylogenetic Trees

Prior to the multiple sequence alignment, the CDS sequences of *PsaM, psb30, ChlB, ChlL, ChlN*, and *RPL21* were converted to amino acid sequences using a sequence manipulation suite (http://www.bioinformatics.org/sms2/translate.html) [41]. The resulting protein sequences were subjected to multiple sequence alignment using the Multalin server to identify conserved amino acid motifs [42]. The CDS sequences of *PsaM, psb30, ChlB, ChlL, ChlN*, and *RPL21* genes were also subjected to multiple sequence alignment using Clustal Omega. The resultant aligned file was downloaded in Clustal format and converted to a MEGA file format using MEGA6 software [43]. The converted MEGA files of *PsaM, psb30, ChlB, ChlL, ChlN*, and *RPL21* were subsequently used for the construction of a phylogenetic tree. Prior to the construction of the phylogenetic tree, a model selection was carried out using MEGA6 software using the following parameters; analysis, model selection; tree to use, automatic (neighbor-joining tree); statistical method, maximum likelihood; substitution type, nucleotide; gaps/missing data treatment, partial deletion; site coverage cut-off (%), 95; branch swap filer, very strong; and codons included, 1st, 2nd, and 3rd. Based on the lowest BIC (Bayesian information criterion) score, the following statistical parameters were used to construct the phylogenetic tree: statistical method, maximum likelihood; test of phylogeny, bootstrap method; number of bootstrap replications, 1000; model/method, general time-reversible model; rates among sites, gamma-distributed with invariant sites (G+I); number. of discrete gamma categories, 5; gaps/missing data treatment, partial deletion; site coverage cut-off (%), 95; ML Heuristic method, nearest-neighbor-interchange (NNI); branch swap filer, very strong; and codons included, 1st, 2nd, and 3rd. The resulting phylogenetic trees were saved as gene trees. Whole-genome sequences of chloroplast genomes were also collectively used to construct a phylogenetic tree to gain insight into the evolution of chloroplast genomes. ClustalW program was used in a Linux-based platform to construct the phylogenetic tree of chloroplast genomes using the neighbor-joining method and 500 bootstrap replicates. The resultant Newick file was uploaded in Archaeopteryx (https://sites.google.com/site/cmzmasek/home/software/archaeopteryx) to view the phylogenetic tree. A separate phylogenetic tree of species with IR-deleted regions was also constructed using the whole sequence of the IR-deleted chloroplast genome using similar parameters as described above. The evolutionary time of plant species used in this study was created using the TimeTree [44]. Cyanobacterial species were used as an outgroup to calibrate the time tree for the other species. 

### 2.3. Analysis of the Deletion and Duplication of Chloroplast-Encoded Genes

A species tree was constructed using the NCBI taxonomy browser (https://www.ncbi.nlm.nih.gov/Taxonomy/CommonTree/wwwcmt.cgi) prior to the study of deletion and duplication of *PsaM, psb30, ChlB, ChlL, ChlN*, and *RPL21* genes. The gene tree of the individual gene family was uploaded in Notung software v.2.9 followed by uploading the species tree and subsequent reconciliation of the gene tree with the species tree [45,46,47]. Once reconciled, deletion and duplication events for the genes were visualized and noted. 

### 2.4. Recombination Events and Time Tree Construction of the Chloroplast Genome 

The constructed phylogenetic tree of chloroplast genomes was uploaded in IcyTree [48] to analyze the recombination events that occurred in chloroplast genomes. The recombination events in IR-deleted and nondeleted IR species were studied separately. The time tree of the studied tree was constructed using the TimeTree program [44]. 

### 2.5. Substitution Rate in Chloroplast Genomes

Chloroplast genomes were grouped into different groups to determine lineage-specific nucleotide substitution rates. The groups were algae, bryophytes, gymnosperms, eudicots, monocots, magnoliids, Nymphaeales, protists, and IR-deleted species. At least 10 chloroplast genomes were included for each lineage when analyzing the rate of nucleotide substitutions. The full-length sequences of chloroplast genomes were subjected to multiple sequence alignment to generate a Clustal file. The MAFT-multiple alignment pipeline was implemented to align the sequences of the different chloroplast genomes. The aligned sequences of individual lineages were downloaded and converted to a MEGA file format using MEGA6 software [43]. The converted files were subsequently uploaded in MEGA6 software to analyze the rate of nucleotide substitution. The following statistical parameters were used to analyze the rate of substitution rate in chloroplast genomes: analysis, estimate transition/transversion bias (MCL); scope, all selected taxa; statistical method, maximum composite likelihood; substitution type, nucleotides; model/method, Tamura–Nei model; and gaps/missing data treatment, complete deletion. 

### 2.6. Statistical Analysis

Principal component analysis and the probability distribution of chloroplast genomes were conducted using Unscrambler software version 7.0 and Venn diagrams were constructed using InteractiVenn (http://www.interactivenn.net/) [49].

## 3. Results

### 3.1. The Genomic Features of Chloroplast Genomes Are Diverse and Dynamic 

A study of 2511 chloroplast genomes was conducted to gain insight into the genomic structure and evolution of the chloroplast genome. The analysis included the complete genome sequences of algae, austrobaileyales, bryophytes, chloranthales, corals, eudicots, Flacourtiaceae, gymnosperms, magnoliids, monocots, Nymphaeales, opisthokonta, protists, pteridophytes, and an unclassified chloroplast genome (Appendix A). A comparison of the analyzed genomes indicated that *Haematococcus lacustris* encoded the largest chloroplast genome, comprising 1.352 Mbs; however, *Pilostyles aethiopica* encoded the smallest chloroplast genome, comprising only 0.01134 Mbs (Figure 1) followed by *Pilostyles hamiltoni* (0.01516 Mb), and *Asarum minus* (0.0155 Mb). The overall average size of the chloroplast genome was found to be 0.152 Mbs. The order of the average size (Mbs) of the chloroplast genome in different plant groups was 0.164 (algae), 0.160 (Nymphaeales), 0.154 (eudicot), 0.154 (Magnoliid), 0.149 (pteridophyte), 0.144 (monocot), 0.134 (bryophyte), 0.131 (gymnosperm), and 0.108 (protist). The average chloroplast genome size in algae (0.164 Mbs) and the Nymphaeales (0.160 Mbs) was larger than eudicots (0.154 Mbs), monocots (0.144 Mbs), and gymnosperms (0.131 Mbs). The average size of the protist chloroplast genome (0.108 Mbs) was the smallest. Principal component analysis (PCA) of the chloroplast genome size of algae, bryophytes, eudicots, gymnosperms, magnoliids, monocots, Nymphaeales, protists, and pteridophytes reveals a clear distinction between the different plant groups (Figure 2). The size of the chloroplast genome of gymnosperm and bryophytes grouped together; and eudicots, magnoliids, and pteridophytes grouped together. In contrast, the algae and protists were independently grouped (Figure 2). This shows that the chloroplast genome of algae and protists might have evolved from their respective common ancestors. 

The number of coding sequences (CDS) in the analyzed chloroplast genomes ranged from 273 (*Pinus koraiensis* and *Choreocolax polysiphoniae*) to 3 (*Pilostyles aethiopica*; Figure 1). The average number of CDS in all the studied chloroplast genome was 89.90 per genome. However, some other species contained a higher number of CDS in the chloroplast genome, including *Grateloupia filicina* (233), *Osmundaria fimbriata* (224), *Porphyridium purpureum* (224), *Lophocladia kuetzingii* (221), *Kuetzingia canaliculata* (218), *Spyridia filamentosa* (218), *Bryothamnion seaforthii* (216) and others (Appendix A). Similarly, some species encoded a lower number of CDS in the chloroplast genome, including *Pilostyles aethiopica* (3), *Pilostyles hamiltoni* (4), *Asarum minus* (8), *Cytinus hypocistis* (15), *Sciaphila densiflora* (18), *Gastrodia elata* (20), *Burmannia oblonga* (22), *Orobanche gracilis* (24), and others (Appendix A). PCA analysis indicated that the number of CDS in bryophytes, eudicots, magnoliids, monocots, and pteridophytes grouped together (Figure 3). The number of CDS in algae, gymnosperms, and protists grouped very distantly from the above-mentioned grouping (Figure 3). The average CDS number in algae (140.93) was quite high compared to magnoliid (84), eudicot (83.55), monocot (82.53), gymnosperm (82.56), and protist (98.97). However, algae and protists encoded a higher number of CDS compared to the magnoliid, eudicot, monocot, gymnosperm, and protist. The larger genome size of algae and protist is associated with a greater number of CDS in the chloroplast genome and they fall distantly in the PCA plot. This suggests that the evolution of chloroplast genome and CDS number of algae and protist share a slightly similar trend compared to other plant species. However, they might have evolved from their respective ancestors. 

The GC content of the analyzed chloroplast genomes ranged from a high of 57.66% (*Trebouxiophyceae* sp. MX-AZ01) to a low of 20.46% (*Choreocolax polysiphoniae*; Figure 4, Appendix A). The average GC content in the chloroplast genome was 36.82%. Some species contained a higher percentage of GC content, including *Trebouxiophyceae* sp. MX-AZ01 (57.664%), *Coccomyxa subellipsoidea* C-169 (50.73%), *Paradoxia multiseta* (50.58%), *Haematococcus lacustris* (49.88%), *Chromerdia* sp. RM11 (47.74%), *Elliptochloris bilobata* (45.76%), *Choricystis parasitica* (45.44%), and others. On the other hand, some species had a lower percentage of GC content, including *Ulva prolifera* (24.78%), *Ulva linza* (24.78%), *Ulva fasciata* (24.86%), *Ulva flexuosa* (24.97%), and others (Appendix A). PCA analysis revealed that the percentage GC content of eudicots, gymnosperms, magnoliids, monocots, and Nymphaeales grouped together, and the percentage of GC content in algae and protists grouped together (Figure 5). The percentage of GC content in bryophytes and pteridophytes did not group with the algae and protists or the eudicots, gymnosperms, magnoliids, monocots, or Nymphaeales (Figure 5). The GC content of algae and protists showed that they have a common trend of evolution with regard to genome size, CDS number, and GC content. The evolutionary similarity of algae and protist is closer than other lineages.

### 3.2. *PsaM, Psb30, ChlB, ChlL, ChlN,* and *RPL21* Are Chloroplast Genes Characteristic of Algae, Bryophytes, Pteridophytes, and Gymnosperms

The PsaM protein is subunit XII of photosystem I. Among the 2511 studied species, 84 were found to possess the *PsaM* gene. All of the species found to possess the *PsaM* gene belonged to algae, bryophytes, pteridophytes, and gymnosperms (Appendix A). Notably, no the species in the angiosperm lineage possessed the *PsaM* gene; clearly indicating that the *PsaM* gene was lost in the angiosperm lineage. The PsaM protein was found to contain the characteristic conserved amino acid motif Q-x_3_-A-x_3_-A-F-x_3_-I-L-A-x_2_-L-G-x_2_-L-Y (Appendix A). A few species, including *Cephalotaxus, Podocarpus tortara, Retrophyllum piresii, Dacrycarpus imbricatus, Glyptostrobus pensilis, T. distichum, Cryptomeria japonica, Pinus contorta, Pinus taeda*, and *Ptilidium pulcherrimum,* however, did not contain the conserved amino acid motif. Instead, they possessed the conserved motif, F-x-S-x_3_-C-F-x_4_-F-S-x_2_-I (Appendix A). Phylogenetic analysis revealed that *PsaM* genes grouped into five independent clusters, suggesting that they have evolved independently from multiple common ancestral nodes (Appendix A). Duplication and deletion analysis of *PsaM* genes revealed that deletion events were more prominent than the duplication or codivergence events (Table 1). Among the 84 analyzed *PsaM* genes, 12 underwent duplication and 34 underwent deletions, while 34 underwent codivergence (Table 1, Appendix A). 

A total of 164 species were found to possess *Psb30* gene and all of the species belonged to algae, bryophytes, pteridophytes, or gymnosperms (Appendix A). *Psb30* was absent in the chloroplast genome of angiosperms. Multiple sequence alignment revealed the presence of a conserved consensus amino acid sequence, N-x-E-x_3_-Q-L-x_2_-L-x_6_-G-P-L-V-I (Appendix A). Phylogenetic analysis of *Psb30* genes resulted in the designation of two major clusters and six minor clusters, suggesting that it evolved from multiple common ancestral nodes (Appendix A). Deletion/duplication analysis indicated that 39 of *Psb30* genes underwent a duplication event and 120 underwent a deletion event, while 49 were found to be codiverged (Table 1, Appendix A). 

*ChlB* encodes a light-independent protochlorophyllide reductase. A total of 288 of the examined chloroplast genome sequences were found to possess a *ChlB* gene (Appendix A) among protists, algae, bryophytes, pteridophytes, and gymnosperms. The *ChlB* gene was absent in species in the chloranthales, corals, or angiosperm lineage. Multiple sequence alignment revealed the presence of several highly conserved amino acid motifs (Appendix A). At least seven conserved motifs were identified, including A-Y-W-M-Y-A, L-P-K-A-W-F, E-N-Y-I-D-Q-Q, S-Q-A-A-W-F, H-D-A-D-W-F, E-P-x_2_-I-F-G-T, E-K-F/Y-A-R-Q-Q, and E-V-M-Y-A-A (Appendix A). Phylogenetic analysis indicated that *ChlB* genes grouped into two major clusters and 13 minor clusters, reflecting multiple evolutionary nodes (Appendix A). *ChlB* genes were composed of a few groups. Specifically, deletion and duplication analysis revealed that 35 *ChlB* genes underwent duplications and 126 underwent deletions, while 116 exhibited codivergence in their evolutionary history (Table 1, Appendix A). 

Analysis of the chloroplast genome sequences identified 303 species that possess *ChlL* genes (Appendix A). All of the identified species possessing the *ChlL* gene belonged to algae, bryophytes, gymnosperms, protists, and pteridophytes. None of the taxa in the angiosperm or magnoliid lineage were found to possess a *ChlL* gene. Within the protist lineage, only species in the genera *Nannochloropsis*, *Vaucheria, Triparma*, and *Alveolata* encode a *ChlL* gene. Multiple sequence alignment revealed the presence of several highly conserved amino acid motifs, including K-S-T-T-S-C-N-x-S, W-P-E-D-V-I-Y-Q, K-Y-V-E-A-C-P-M-P, C-D-F-Y-L-N, Q-P-E-G-V-V/I, and S-D-F-Y-L-N (Appendix A). The phylogenetic analysis indicated that *ChlL* genes grouped into one major independent cluster and 11 minor clusters, suggesting that they also evolved independently from different common ancestors (Appendix A). Deletion and duplication analysis indicated that 49 *ChlL* genes underwent duplication events and 184 underwent deletions, while 100 *ChlL* genes exhibited codivergence (Table 1, Appendix A). 

The analysis revealed that at least 289 species possess *ChlN* genes. These genomes were from taxa within the protists, algae, bryophytes, pteridophytes, and gymnosperms (Appendix A). Multiple sequence alignment revealed the presence of highly conserved amino acid motifs, including N-Y-H-T-F, A-E-L-Y-Q-K-I-E-D-S, M-A-H-R-C-P, and Q-I-H-G-F (Appendix A). Phylogenetic analysis revealed that *ChlN* genes group into two independent clusters (Appendix A). No lineage-specific grouping, however, was identified in the phylogenetic tree. Deletion and duplication analysis indicated that eight *ChlN* genes underwent duplication events, 46 underwent deletion events and 34 genes exhibited codivergence (Table 1, Appendix A). 

The chloroplast genomes of at least 137 of the examined species were found to possess an *RpL21* gene which belonged to algae, bryophytes, pteridophytes, and gymnosperms (Appendix A). In the majority of cases, full-length CDS was not found. Instead, the CDS of the *Rpl21* genes were found to be truncated. Therefore, only 22 full-length CDS were used to identify deletion and duplication events. Rpl21 proteins were found to contain the conserved amino acid motifs, Y-A-I-I-D-x-G-G-x-Q-L-R-E-V-x-G-R-F, R-V-L-M-I, G-x-P-W-L, R-I-L-H, and K-x_2_-I/V-x_5_-K-K (Appendix A). Phylogenetic analysis shows the presence of three clusters, reflecting their origin from multiple common ancestral nodes (Appendix A). Deletion/duplication analysis indicated that three *RpL21* genes underwent duplication events, eight underwent deletion events, and nine exhibited codivergence (Table 1, Appendix A).

### 3.3. The *Rbcl* Gene Has Been Lost in Parasitic and Heterotrophic Plant Species 

The analysis found, at least 19 species have lost the *Rbcl* gene in their chloroplast genome. The species lacking an *Rbcl* gene were *Pilostyles aethiopica*, *Pilostyles hamiltoni*, *Alveolata* sp. CCMP3155, *A. minus, Bathycoccus prasinos* (picoplankton)*, Burmannia oblonga* (orchid)*, Codonopsis lanceolate* (eudicot)*, Cytinus hypocistis* (parasite)*, Gastrodia elata* (saprophyte)*, Monotropa hypopitys* (mycoheterotroph)*, Orobanche austrohispanica* (parasite)*, Orobanche densiflora* (parasite)*, Orobanche gracilis* (parasite)*, Orobanche pancicii* (parasite)*, Phelipanche purpurea* (parasite)*, Phelipanche ramosa* (parasite)*, Prototheca cutis* (parasitic algae)*, Prototheca stagnorum* (parasitic algae)*,* and *Sciaphila densiflora* (mycoheterotroph). 

### 3.4. Deletion of Inverted Repeats (IRs) Has Occurred across All Plastid Lineages 

Inverted repeats (IR) are one of the major characteristic features of the chloroplast genomes. The analysis conducted in the present study revealed the deletion of inverted repeats in the chloroplast genome of 259 (10.31%) species from the 2511 species examined (Appendix A). IR deletion events were identified in protists (14), protozoans (one), algae (126), bryophytes (one), gymnosperms (64), magnoliids (one) monocots (nine), and eudicots (43). The average size of the deleted IR region in algae was 0.177 Mb, which is larger than the overall size of the chloroplast genome in the respective taxa. The average size of the deleted IR region in eudicots, monocots, and gymnosperms was 0.124, 0.131, and 0.127 Mb, respectively, which is smaller than the overall size of the chloroplast genome in the respective lineages.

Phylogenetic analysis of chloroplast genomes containing deleted IR regions produced three major clusters (Appendix A). Gymnosperms were in the upper cluster (cyan) while the lower cluster (red) comprised the algae, bryophytes, eudicots, gymnosperms, and pteridophytes. No chloroplast genomes from monocot plants were present in the lower cluster (Appendix A). The middle cluster contained at least four major phylogenetic groups (Appendix A). Monocot plants were present in two groups (pink) in the middle cluster. Gymnosperm (cyan) and eudicot (green) chloroplast genomes were also present in two of the groups in the middle cluster. Although there was some sporadic distribution of algae in different groups of the phylogenetic tree, the majority of the algal species were present in a single group (yellow; Appendix A). A phylogenetic tree of taxa with deleted IR and taxa with chloroplast genomes that did not lose the IR region (*Floydiella terrestris*, *Carteria cerasiformis, B. apyrenoidosa, Eucalyptus grandis, Oryza sativa,* and others) did not reveal any specific difference in their clades. Instead, they also grouped with the genomes in which the IR region was deleted. Inverted repeats stabilize the chloroplast genome [50,51] and the loss of a region of inverted repeats most likely leads to a genetic rearrangement in the chloroplast genome. The lower cluster (red) contained the oldest group. Genomic recombination analysis revealed that the chloroplast genomes across different lineages also underwent vivid recombination (Appendix A). In addition, the IR-deleted chloroplast genomes also underwent vivid recombination (Appendix A). 

### 3.5. Several Genes in the Chloroplast Genome Have Been Lost

The chloroplast genome encodes genes for photosynthesis, amino acid biosynthesis, transcription, protein translation, and other important metabolic processes. The major genes involved in such events are *AccD* (acetyl-coenzyme A carboxylase carboxyl transferase), *AtpA, AtpB, AtpE, AtpF, AtpH, AtpI, CcsA* (cytochrome C biogenesis protein), *CemA* (chloroplast envelope membrane), *ChlB* (light-independent protochlorophyllide reductase), *ChlL, ChlN, ClpP* (ATP-dependent Clp protease), *MatK* (maturase K), *NdhA* (NADPH-quinone oxidoreductase), *NdhB, NdhC, NdhD, NdhE, NdhF, NdhG, NdhH, NdhI, NdhJ, NdhK, Pbf1* (photosystem biogenesis factor 1), *PetA* (cytochrome precursor), *PetB, PetD, PetG, PetL, PetN, PsaA* (photosystem I protein), *PsaB, PsaC, PsaI, PsaJ, PsaM, Psb30, PsbA* (photosystem II protein), *PsbB, PsbC, PsbD, PsbE, PsbF, PsbH, PsbI, PsbJ, PsbK, PsbL, PsbM, PsbT, PsbZ, Rbcl* (ribulose 1,5-bisphosphate carboxylase), *Rpl2* (60S ribosomal protein), *Rpl14, Rpl16, Rpl20, Rpl21, Rpl22, Rpl23, Rpl32, Rpl33, Rpl36, RpoA* (DNA-directed RNA polymerase), *RpoB, RpoC1, RpoC2, Rps2 (40S ribosomal protein), Rps3, Rps4, Rps7, Rps8, Rps11, Rps12, Rps14, Rps15, Rps16, Rps18, Rps19, Ycf1, Ycf2, Ycf3, and Ycf4*. Our analysis revealed that a number of these genes were lost in one or other species in a dynamic manner (Table 2). The analysis indicated that the ribosomal proteins Rpl and Rpo were lost less frequently than the other chloroplast genes (Table 2). *Ndh* genes were lost in a number of different species. Several other genes had been deleted in a considerable number of species across different lineages. These included *AccD* (402), *AtpF* (217), *Clp* (194), *Ycf2* (226), *Ycf4* (111), *PetL* (248), *PetN* (125), *PsaI* (129), *PsbM* (166), *PsbZ* (145), *Rpl22* (137), *Rpl23* (221), *Rpl32* (182), *Rpl33* (163), *Rps15* (263), and *Rps16* (372), where the number in parentheses indicates the number of taxa in which the gene has been deleted from the chloroplast genome (Table 2). Detailed about the loss of all the chloroplast genes across can be found in Appendix A. 

### 3.6. The Loss of Genes in Chloroplast Genomes is Dynamic 

When the collection of all the lost genes were grouped, it was evident that a large number of genes had been found to be lost in algae, eudicots, magnoliids, and monocots (Appendix A). Only a small number of genes were lost in bryophytes, gymnosperms, protists, and pteridophytes (Appendix A). When the species of algae, gymnosperms, monocots, eudicots, magnoliids, and bryophytes were grouped together, *NdhA, NdhC, NdhD, NdhE, NdhF, NdhG, NdhH, NdhI, NdhJ,* and *NdhK* genes were found to be lost in all six lineages; however, *AtpB, AtpE, AtpH, AtpI, CemA, PetA, PetB, PetD, PetG, PetL, PsaA, PsaB, PsaC, PsaI, PsbA, PsbB, PsbC, PsbD, PsbE, PsbF, PsbH, PsbJ, PsbL, PsbZ, Psbf1, Rpl22, Rpl33, RpoB,* and *RpoC2* had been lost in algae, eudicots, magnoliids, and monocots (Appendix A, Appendix A). *AccD, NdhB, PsaJ, Rpl23*, and *Rpl32* genes were only absent in species of algae, eudicots, gymnosperms, magnoliids, and monocots. When species of algae, bryophytes, gymnosperms, angiosperms (monocot and dicot), pteridophytes, and protists were grouped together, at least 11 genes were found to be lost in all of the lineages (Appendix A, Appendix A). The most commonly lost genes were *NdhA, NdhC, NdhD, NdhE, NdhF, NdhG, NdhH, NdhI, NdhJ, NdhK*, and *Rps16*. The *NdhB* gene, however, was lost in algae, angiosperms, gymnosperms, protists, and pteridophytes; however, it was present in all species of bryophytes. When the higher groupings of plant lineages (gymnosperms, magnoliids, and monocots) were grouped together, it was found that *AccD, NdhA, NdhB, NdhC, NdhD, NdhE, NdhF, NdhG, NdhH, NdhI, NdhJ, NdhK, PsaJ, Rpl23*, and *Rpl32* had been lost in all four lineages (Appendix A, Appendix A). *AtpB, AtpE, AtpH, AtpI, CcsA, CemA, PetA, PetB, PetD, PetG, PetL, PetN, PsaA, PsaB, PsaC, PsaI, PsbA, PsbB, PsbC, PsbD, PsbE, PsbF, PsbH, PsbJ, PsbL, PsbZ, Psbf1, Rpl22, Rpl33, RpoB, RpoC1, RpoC2*, and *Rps19* were found to be lost in eudicots, magnoliids, and monocots. *ClpP* was found to be lost in eudicots, gymnosperms, and magnoliids. A comparative analysis of gene loss in eudicot and monocot plants revealed that gene loss was more frequent in eudicots (69 genes) than in monocots (59 genes). Eudicots and monocots share the loss of 59 genes in their chloroplast genomes. The loss of *ClpP, Rpl2, Rpl14, Rpl36, RpoA, Rps2, Rps8, Rps11, Rps14*, and *Rps18* occurred only in eudicots and not in monocots. A comparative analysis of gene loss in eudicots, gymnosperms, and monocots indicated that the loss of *Rps7* was unique to the gymnosperms. The loss of at least 17 genes (*accD, ndhA, ndhB, ndhC, ndhD, ndhE, ndhF, ndhG, ndhH, ndhI, ndhJ, ndhK, psaJ, rpl23, rpl32, rps15,* and *rps16*) were found to be common in between eudicots, gymnosperms, and monocots. 

### 3.7. Chloroplast-Derived Genes Are Present in the Nuclear Genome

It has been speculated that genes lost from chloroplast genomes may have moved to the nuclear genome and are regulated as a nuclear-encoded gene [52,53]. Therefore, a genome-wide analysis of fully sequenced and annotated genomes of 145 plant species was analyzed to explore this question. Results indicated a maximum presence of the chloroplast-encoding genes in the nuclear genome. We found the presence of 189,381 putative nuclear encoding chloroplast gene from the study of 145 plant species (Appendix A). Some of the chloroplast-derived genes that were found in the nuclear genome were: Rubisco accumulation factor, 30S ribosomal 30S ribosomal proteins (1, 2, 3, S1, S2, S3, S5, S6, S7, S8, S9, S10, S11, S12, S13, S14, S15, S16, S17, S18, S19, S20, S21, and S31) 50S ribosomal proteins (5, 6, L1, L2, L3, L4, L5, L6, L9, L10, L11, L12, L13, L14, L15, L16, L17, L18, L19, L20, L21, L22, L23, L24, L27, L28, L29, L31, and L32), *Psa* (A, B, C, I, and J), *Psb* (A, B, D, E, F, H, I, J, K, L, M, N, P, Q, T, and Z), *Rpl* (12 and 23), *RpoA*, *RpoB*, *RpoC1, RpoC2, Rps7, Rps12, Ycf* (1, 2, and 15), *YlmG* homolog, Ribulose bisphosphate carboxylase small chain (1A, 1B, 2A, 3A, 3B, 4, F1, PW9, PWS4, and S4SSU11A), Ribulose bisphosphate carboxylase/oxygenase activase A and B, (-)-beta-pinene synthase, (-)-camphene/tricyclene synthase, (+)-larreatricin hydroxylase, (3S,6E)-nerolidol synthase, (E)-beta-ocimene synthase, 1,4-alpha-glucan-branching enzyme, 10 kDa chaperonin, 1,8-cineole synthase, 2-carboxy-1,4-naphthoquinone phytyltransferase, 2-C-methyl-D-erythritol 2,4-cyclodiphosphate synthase, 2-C-methyl-D-erythritol 4-phosphate cytidylyltransferase, ABC transporter B family, AccD, acyl-carrier-protein, adenylate kinase, ALBINO protein, allene oxide cyclase, anion transporter, anthranilate synthase, APO protein, aspartokinase, ATP synthase, Atp (A, B, E, F, H, I), ATP-dependent Clp protease, beta carbonic anhydrase, calcium-transporting ATPase, Calvin cycle protein CP12, carbonic anhydrase, cation/H(+) antiporter, chaperone protein Clp (B, C, and D), DnaJ, chaperonin 60 subunit, chlorophyll a-b binding protein (1, 2, 3, 4, 6, 7, 8, 13, 15, 16, 21, 24, 26, 29, 36, 37, 40, 50, 80, M9, LHCII, and P4), chlorophyll(ide) b reductase (NOL and NYC), chloroplastic acetyl-coenzyme A carboxylase, chloroplastic group IIA intron splicing facilitator CRS (S1, A, and B), chorismate mutase, cytochrome b6/f complex subunit (1, 2, IV, V, VI, and VIII), cytochrome c biogenesis protein CCS1, DEAD-box ATP-dependent RNA helicase, DNA gyrase A and B, DNA polymerase A and B, DNA repair protein recA homolog, DNA-(apurinic or apyrimidinic site) lyase, DNA-damage-repair/toleration protein, DNA-directed RNA polymerase, early light-induced protein, fatty acid desaturase, ferredoxin--NADP reductase, fructokinase, gamma-terpinene synthase, geraniol synthase, geranylgeranyl pyrophosphate synthase, glucose-1-phosphate adenylyltransferase small and large subunit, glutathione S-transferase, GTP diphosphokinase CRSH, inactive ATP-dependent zinc metalloprotease FTSHI, inactive shikimate kinase, kinesin protein KIN (D, E, K, L, and M), L-ascorbate peroxidase, light-harvesting complex protein, light-induced protein, light-regulated protein, lipoxygenase, magnesium transporter, magnesium-chelatase, MATE efflux family protein, multiple organellar RNA editing factor, N-(5′-phosphoribosyl)anthranilate isomerase, NAD Kinase, NAD(P)H-quinone oxidoreductase subunits (1, 2, 3, 4, 5, 6, H, I, J, K, L, M, N, O, S, T, and U), NADH dehydrogenase subunits (1, 2, 3, 4, 5, 6, 7, I, J, and K), NADH-plastoquinone oxidoreductase subunits (1, 2, 3, 4, 5, 6, 7, I, J, and K), NADPH-dependent aldehyde reductase, nifU protein, nudix hydrolases, outer envelope pore proteins, oxygen-evolving enhancer proteins, pentatricopeptide repeat-containing protein (CRP1, DOT4, DWY1, ELI1, MRL1, OTP51, PPR5), peptide chain release factor, peptide methionine sulfoxide reductase, peptidyl-prolyl cis-trans isomerases, Pet (A, B, G, and L), phospholipase, photosynthetic NDH subunit of lumenal location, photosynthetic NDH subunit of subcomplex B, protochlorophyllide reductase subunits (B, L, and N), phytol kinase, plastid-lipid-associated proteins, protease Do 1, protein cofactor assembly of complex c subunits, protein CutA, DCL, pyruvate dehydrogenase E1 component subunits, sodium/metabolite cotransporter BASS, soluble starch synthase, stearoyl-[acyl-carrier-protein] 9-desaturase, thioredoxins, thylakoid luminal proteins, translation initiation factor, transcription factor GTE3, transcription termination factor MTERF, translocase of chloroplast, zinc metalloprotease EGY, and others (Appendix A).

### 3.8. The Ratio of Nucleotide Substitution Is Highest in Pteridophytes and Lowest in Nymphaeales 

Determining the rate of nucleotide substitution in the chloroplast genome can be an important parameter that needs to be more precisely understood to further elucidate the evolution of the chloroplast genome. Single base substitutions, and insertion and deletion (indels) events play an important role in shaping the genome. Therefore, an analysis was conducted to determine the rate of substitution in the chloroplast genome by grouping them according to their respective lineages. Results indicated that the transition/transversion substitution ratio was highest in pteridophytes (k1 = 4.798 and k2 = 4.043) and lowest in Nymphaeales (k1 = 2.799 and k2 = 2.713; Appendix A). The ratio of nucleotide substitution in species with deleted IR regions was 2.951 (k1) and 3.42 (k2; Appendix A). The rate of transition of A > G substitution was highest in pteridophytes (15.08) and lowest in protists (8.51) and the rate of G > A substitution was highest in protists (22.15) and lowest in species with deleted IR regions (16.8). The rate of substitution of T > C was highest in pteridophytes (14.01) and lowest in protists (8.95; Appendix A). The rate of substitution of C > T was highest in protists (22.34) and lowest in Nymphaeales. The rate of transversion is two-times less frequent than the rate of transition. The rate of transversion of A > T was highest in protists (6.80) and lowest in pteridophytes (4.64), while the rate of transversion of T > A was highest in algae (6.98) and lowest in pteridophytes (Appendix A). The rate of substitution of G > C was highest in Nymphaeales (4.31) and lowest in protists (2.46), while the rate of substitution of C > G was highest in Nymphaeales (4.14) and lowest in protists (2.64; Appendix A). Based on these results, it is concluded that the highest rates of transition and transversion were more frequent in lower eukaryotic species, including algae, protists, Nymphaeales, and pteridophytes; however, high rates of transition/transversion were not observed in bryophytes, gymnosperms, monocots, and dicots (Appendix A). Notably, G > A transitions were more prominent in chloroplast genomes with deleted IR regions (Appendix A).

### 3.9. Chloroplast Genomes Have Evolved from Multiple Common Ancestral Nodes

A phylogenetic tree was constructed to obtain an evolutionary perspective of chloroplast genomes (Figure 6). All of the 2511 studied species were used to construct a phylogenetic tree (Figure 6). The phylogenetic analysis produced four distinct clusters, indicating that chloroplast genomes evolved independently from multiple common ancestral nodes. Lineage-specific groupings of chloroplast genomes were not present in the phylogenetic tree. The genomes of algae, bryophytes, gymnosperms, eudicots, magnoliids, monocots, and protists grouped dynamically in different clusters. Although the size of the chloroplast genome in protists was far smaller than other lineages and still, they were distributed sporadically throughout the phylogenetic tree. Time tree analysis indicated that the origin of the cyanobacterial species (used as outgroup) date back to ~2180 Ma and that the endosymbiosis of the cyanobacterial genome occurred ~1768 Ma ago and was incorporated into the algal lineage ~1293-686 Ma ago (Appendix A); which then further evolved into the Viridiplantae ~1160 Ma, Streptophyta ~1150 Ma, Embryophyta ~532 Ma, Tracheophyte ~431 Ma, Euphyllophyte 402 Ma, and Spermatophyta 313 Ma (Appendix A). The molecular signature genes *PsaM, ChlB, ChlL, ChlN, Psb30*, and *Rpl21* in algae, bryophytes, pteridophytes, and gymnosperms were lost ~203 (Cycadales) and -156 (Gnetidae) Ma ago, and as a result, are not found in the subsequently evolved angiosperm lineage (Appendix A).

## 4. Discussion

Chloroplasts are an indispensable part of plant cells function as semiautonomous organelles due to the presence of their own genetic material, potential to self-replicate, and capability to modulate cell metabolism [4,54,55,56]. The size of the chloroplast genome is highly variable and does not correlate to the size of the corresponding nuclear genome of the species. The average size of the chloroplast genome is 0.152 Mb and encodes an average of 91.67 CDS per genome. The deletion of IR regions in the chloroplast genome is supposed to drastically reduce the genetic content of the chloroplast genome and also the number of CDS. However, the current analysis does not support this premise. The average number of CDS in algae (140.93) was higher than protists (98.97), pteridophytes (86.54), eudicots (83.55), bryophytes (83.38), gymnosperms (82.54), and monocots (82.53). The larger genome size (0.177 Mb) of the chloroplast genome in algae with deleted IR regions, and the higher number of CDS (172.16 per genome) in IR-deleted taxa of algae indicates that the loss of IR regions in algae led to a genetic rearrangement and an enlargement in the chloroplast genome. However, the average CDS number of other lineages in IR-deleted genomes was quite lower than their average CDS count (86.28 for protist, 63 for monocot, 81.42 for gymnosperm, and 71.88 for eudicot). The average size of IR-deleted chloroplast genomes in eudicots, monocots, protists, and gymnosperms was smaller than the average size of chloroplast genomes of taxa where IR regions have not been deleted. Thus, the lower number of CDS in these taxa may be related to the deletion of IR regions. This suggests that the deletion of IR regions in the chloroplast genome of algae is directly proportional to the increase in the genome size and concomitant increase in the CDS number; however, this was not true in the other plant lineages where the relationship was inversely proportional. The deletion of IR regions has been previously reported in a few species of algae, magnoliids, and other genomes [57,58,59,60,61]. The present study, however, provided clear evidence regarding the loss of IR regions across all plant and protist lineages. The deletion of IR repeats and an increase in the genome size in algae has largely been attributed to the duplication of the chloroplast genome. The evolutionary age of IR-deleted species of algae dates back to ~965-850 Ma. This provides strong evidence that the deletion of IR repeats and duplications of the chloroplast genome has been a continuous process since the initial evolution of the chloroplast genome in algae. Zhu et al. also suggested a role for duplication in the evolution of IR-deleted chloroplast genomes [60]. Characterizing the pattern and frequency of neutral mutations (substitution, insertions, and deletion) is important for deciphering the molecular basis of the evolution of genes and genomes. Turmel et al. reported that a differential loss of genes from the chloroplast genome resulted in the loss of IR regions in the chloroplast genome for all the lineages, except algae and protists [57]. The transition/transversion ratio of purine substitutions in all IR-deleted species (k1 = 2.951) was much lower than in non-IR-deleted species, except for species in the Nymphaeales, and the substitution of pyrimidines in all IR-deleted species was higher (k2 = 3.42), except pteridophytes (Appendix A). These data suggest that, in addition to a duplication event, a lower rate of purine substitution and a higher rate of pyrimidine substitution are closely associated with the deletion of IR regions. 

In addition to the loss of IR regions, the loss of genes from chloroplast genomes was also analyzed. The loss of important genes from the chloroplast genome has been previously reported in some species of green algae, bryophytes, and magnoliids (Appendix A) [62,63,64,65]. The results of the present study indicate the loss of the *Rbcl* gene in at least 19 species among parasitic, mycoparasitic, and saprophytic plant species across different lineages, including algae, eudicots, magnoliids, monocots, and protists. The parasitic plant *Conopholis* of Orobanchacea lost the photosynthetic gene *Rbcl*; however, it was present in other parasitic plants in Orobanchacea [66,67]. The loss of *Rbcl*, however, was not observed in any species of bryophytes, pteridophytes, or gymnosperms. The number of CDS in the *Rbcl*-deleted chloroplast genome was much lower (27 per genome) relative to the average number of CDS found in the chloroplast genomes; except for *Alveolata* sp. CCMP3155 which possessed 81 CDS. The loss of the *Rbcl* gene in the chloroplast genome is associated with a drastic reduction in the number of other protein-coding genes. The reduction in the genome size is associated with the massive loss of ancestral protein-coding genes [68]. Interestingly, the parasitic genus, *Cuscuta,* possesses an *Rbcl* gene which suggests that the parasitic nature of a species is not always associated with the deletion of the *Rbcl* gene and vice versa, the loss of the *Rbcl* gene is not a prerequisite of becoming a parasitic plant as well. However, it is quite clear that parasitism is getting more prone towards the loss of chloroplast-encoding genes. Although a few contain the *Rbcl* gene, they cannot sustain themselves for their own photosynthesis. The losses of these molecular features are providing an important platform to understand the plant–parasite interactions and evolution of parasitic plants. The loss of genes is most possibly associated with a high level of contraction of the nuclear genome as well. Most possibly, the autotrophic plant evolved parasitic characters through neofunctionalization and transcriptional reprogramming of its older lineage. The study reported that transition from the autotrophic plants to parasitic plants relaxes the functional constraints in a stepwise manner for plastid genes [69].

The deletions of one or more important genes of the chloroplast genome observed in numerous species (Appendix A). It is difficult to decipher the exact reason for the loss of these individual genes in different chloroplast genomes. *NdhA, NdhC, NdhD, NdhE, NdhF, NdhG, NdhH, NdhI, NdhJ, NdhK,* and *Rps16* were genes that were most commonly lost across the analyzed chloroplast genomes. The *NdhB* gene, however, was found to be intact in all species of bryophytes, suggesting that it could serve as a signature gene for the bryophyte chloroplast genome. *Ndh* genes encode a component of the thylakoid Ndh-complex involved in photosynthetic electron transport. The loss of specific *Ndh* genes in different species suggests that not all *Ndh* genes are involved in or needed for functional photosynthetic electron transport. The loss of one *Ndh* gene may be compensated for by other *Ndh* genes or by nuclear-encoded genes. The functional role of the *Ndh* gene was previously reported to be closely related to the adaptation of land plants and photosynthesis [70]. The loss of *Ndh* genes in species across all the plant lineages, including algae, suggests that *Ndh* genes are not associated with the adaptation of photosynthesis to terrestrial ecosystems. Previous studies have reported the loss of *Ndh* genes in the Orchidaceae, where the deletion was reported to occur independently after the orchid family split into different subfamilies [71]. These data suggest that the loss of *Ndh* genes in the parental lineage of orchids led to the loss of *Ndh* genes in the subfamilies in the downstream lineages of orchids. 

A comparison of gene loss in monocots and dicots revealed that species in the eudicots are more prone to gene loss than monocot species. Monocots and dicots chloroplast genome shared a common loss of 59 genes, while eudicots have lost 10 more genes (*ClpP, Rpl14, Rpl2, Rpl36, RpoA, Rps2, Rps8, Rps11, Rps14,* and *Rps18*) than monocots, suggesting that these genes represent the molecular signature of the chloroplast genomes of monocot species. *Ycf* (*Ycf1, Ycf2, Ycf3,* and *Ycf4*) genes were found to be intact in all species of bryophytes, gymnosperms, and pteridophytes, suggesting that they represent a common molecular signature for these lineages. Various genes, including *MatK, Rbcl, Ndh,* and *Ycf*, are commonly used as universal molecular markers in DNA barcoding studies for determining the genus and species of the plants. The loss of these genes in the chloroplast genome of various lineages makes their use as universal markers questionable in future studies for DNA barcoding [72,73,74,75,76]. 

The loss of *RpoA* from the chloroplast genome of mosses was previously reported and it was suggested that *RpoA* had relocated to the nuclear genome [63,77]. The loss of *Psa* and *Psb* genes were quite prominent in algae, eudicot, magnoliid, monocot, and protist lineages. *Psa* and *Psb* genes were always found in species of bryophytes, pteridophytes, and gymnosperms, suggesting that these genes could serve as a common molecular signature for these lineages. *PsaM, Psb30, ChlB, ChlL, ChlN,* and *Rpl21* are characteristic molecular signature genes for lower eukaryotic plants, including algae, bryophytes, pteridophytes, and gymnosperms. Additionally, these genes are completely absent in the eudicots, magnoliids, monocots, and protists. The absence of these genes in angiosperm and magnoliid lineages reflect their potential role in the origin of flowering plants. Duplication events for *PsaM, Psb30, ChlB, ChlL, ChlN*, and *Rpl21* genes were much lower than deletion and codivergence events (Table 1). In fact, codivergence was the dominant event for all of these genes (Table 1). The recombination events that occurred in the chloroplast genome directly reflect the potential possibility of codivergent and divergent evolution in these genes. The presence of *PsaM, Psb30, ChlB, ChlL,* and *ChlN* genes in their respective lineages support the premise that these genes are orthologous and resulted from a speciation event [78,79,80,81]. *Chl* genes are involved in photosynthesis in cyanobacteria, algae, pteridophytes, and conifers [82,83,84,85,86,87]; indicating that the *Chl* genes were originated at least ~2180 Ma ago and remained intact up to the divergence of the angiosperms at ~156 Ma. The loss of *Psa* and *Psb* genes in different species also suggests that they are not essential for a complete and functional photosynthetic process. The loss of a *Psa* or *Psb* gene in a species might be compensated for by other *Psa* or *Psb* genes or by a nuclear-encoded gene. The loss of *Psa* and *Psb* genes in species across all plant lineages has not been previously reported. Thus, this study is the first to report the loss of *Psa* and *Psb* genes in the chloroplast genome of species across all plant lineages, as well as protists. The loss of *Rpl22, Rpl32,* and *Rpl33* genes was more prominent than the loss of *Rpl2, Rpl14, Rpl16, Rpl20, Rpl23,* and *Rpl36*, suggesting the conserved nature of *Rpl2, Rpl14, Rpl16, Rpl20, Rpl23,* and *Rpl36* genes and the conserved transfer of these genes to subsequent downstream lineages as intact genes. *Rpl20* was found to be an intact gene in all 2511 of the studied species, suggesting that *Rpl20* is the most evolutionary conserved gene in the chloroplast genome of the plants and protists. Therefore, *Rpl20* can be considered as the molecular signature gene of the chloroplast genome. Similarly, the loss of *Rps15* and *Rps16* was more frequently relative to the loss of *Rps2, Rps3, Rps4, Rps7, Rps8, Rps11, Rps12, Rps14, Rps18,* and *Rps19*.

There are several reports regarding the transfer of genes from the chloroplast to the nucleus [4,31,88,89,90]. In the present study, almost all of the genes encoded by the chloroplast genomes were also found in the nuclear genome. The presence of the chloroplast-encoded genes in the nuclear genome, however, was quite dynamic. If a specific chloroplast-encoded gene was found in the nuclear genome of one species, it may not have been present in the nuclear genome of the other species. One report also indicated that genes transferred to the nuclear genome may not provide a one to one correspondence function [90]. The question also arises as to how almost all of the chloroplast-encoded genes can be found in the nuclear genome and how were they transferred? If the transfers and correspondence are real, it is plausible that almost all chloroplast-encoded genes have been transferred to the nuclear genome in one or more species and that the transfer of chloroplast genes to the nuclear genome is a common process in the plant kingdom and exchange of chloroplast genes with nuclear genome have already completed. 

## 5. Conclusions

The underlying exact mechanism regarding the deletion of IR regions from the chloroplast genome is still unknown and the loss of specific chloroplast-encoded genes and IR regions in diverse lineages makes it more problematic to decipher the mechanism or selective advantage behind the loss of the genes and IR regions. It is likely that nucleotide substitutions and the dynamic recombination of chloroplast genomes are the factors that are most responsible for the loss of genes and IR regions. Although the evolution of parasitic plants can, to some extent, be attributed to the loss of important chloroplast genes (including *Rbcl*); still it is not possible to draw any definitive conclusions regarding the loss of genes and IR regions. The presence of all chloroplast-encoded genes in the nuclear genome in one or another species is quite intriguing. A question arises, however: do the chloroplast genomes complete the transfer of different chloroplast-encoding genes in different species based on some adaptive requirement? The presence of a completely intact *Rpl20* gene without any deletions in the chloroplast genome of all the species indicates that the *Rpl20* gene can be considered as a molecular signature gene of the chloroplast genome.

## Figures and Tables

**Figure 1 genes-11-01133-f001:**
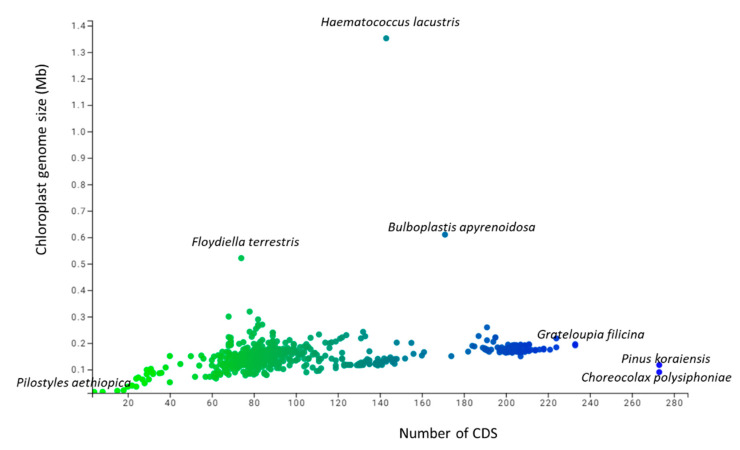
Genome size and number of coding sequences (CDS) in the chloroplast genome. The blue dot present at the right side indicates the genome size of the largest chloroplast genome that encodes 1.35 Mbs in *Haematococcus lacustris* and the green dot present at the top of the figure represents 273 CDS found in *Pinus koraiensis*.

**Figure 2 genes-11-01133-f002:**
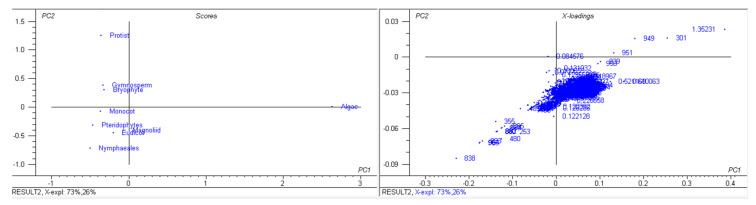
Principal component analysis of chloroplast genome sizes. The genome size of gymnosperms and bryophytes fall in one group and eudicots, magnoliids, monocots, and pteridophytes fall in the other group; however, algae and protists fall distantly.

**Figure 3 genes-11-01133-f003:**
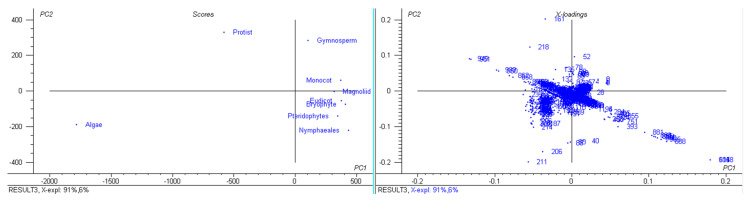
Principal component analysis of CDS numbers of chloroplast genomes. The CDS number of algae, gymnosperms, and protists fall separately; however, bryophytes, eudicots, pteridophytes, and Nymphaeales fall together.

**Figure 4 genes-11-01133-f004:**
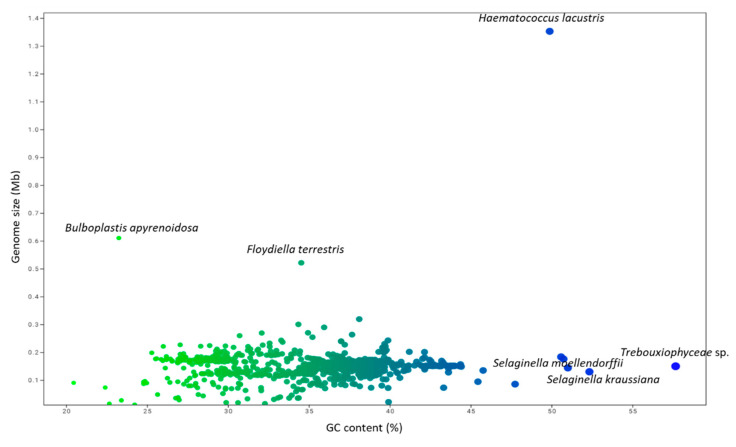
Genome size and GC (%) content in the chloroplast genome. The genome size of *Haematococcus lacustris* was highest (1.352 Mb) present in the upper right side (blue dot). The blue dot present at the right side of the figure represents the GC content of *Trebouxiophyceae* sp. MX-AZ01 that contain 57.66% GC nucleotides; however, the green dot present at the left upper part of the figure represents the lower GC content (23.25%) of *Bulboplastis apyrenoidosa*.

**Figure 5 genes-11-01133-f005:**
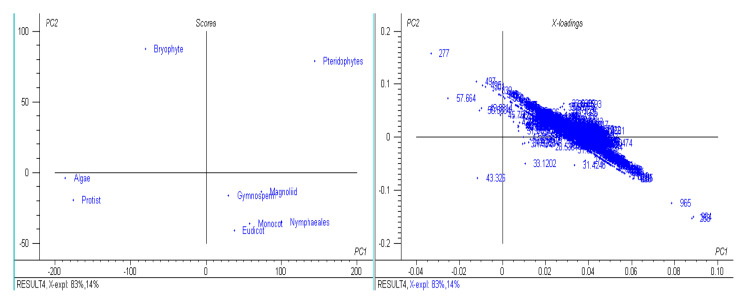
Principal component analysis of GC content of the chloroplast genomes. The GC content of algae and protists and gymnosperms, magnoliids, monocots, eudicots, and Nymphaeales grouped together; however, the GC content of the bryophytes and pteridophytes fall distantly.

**Figure 6 genes-11-01133-f006:**
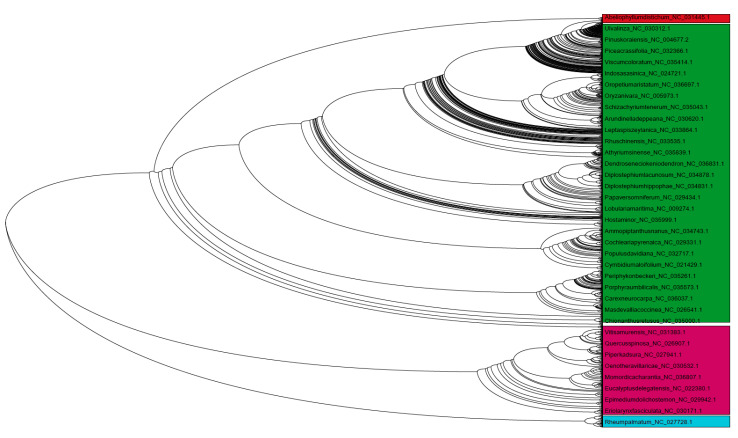
Phylogenetic tree of chloroplast genomes. The phylogenetic tree showed the presence of four major clusters in the chloroplast genomes, suggesting their evolution from multiple common ancestral nodes. The phylogenetic tree considered all of the genomes used during the study and was constructed by a Neighbor-joining program with 500 bootstrap replicates and ClustalW.

**Table 1 genes-11-01133-t001:** Deletion and duplication events of *PsaM, Psb30, ChlB, ChlL, ChlN*, and *Rpl21* genes. Analysis revealed gene loss was dominated compared to the duplication and codivergence.

Name of the Gene	Total No. of Sequences Studied	No. of Duplication	No. of Codivergence	No. of Losses	Transfer
*PsaM*	84	12 (14.28%)	37 (44.04%)	34 (40.47%)	0
*Psb30*	157	39 (24.84)	49 (31.21%)	120 (76.43%)	0
*ChlB*	288	35 (12.15%)	116 (40.27%)	126 (43.75%)	0
*ChlL*	283	49 (17.31%)	100 (35.33%)	184 (65.01%)	0
*ChlN*	83	8 (9.63%)	34 (40.47%)	46 (55.42%)	0
*Rpl21*	22	3 (13.63%)	9 (40.90%)	8 (36.36%)	0

**Table 2 genes-11-01133-t002:** Deletion of different genes in the chloroplast genomes. Almost all of the genes have been deleted in the chloroplast genome of one or another species. However, *Rpl20* was found to be the most intact gene and found in all the species studied so far.

rpoA	rpoB	rpoC1	rpoC2												
26	19	21	13												
atpA	atpB	atpE	atpF	atpH	atpI										
8	8	12	14	13	12										
accD	ccsA	cemA	chlB	chlL	ChlN										
387	29	29	2054	2062	2066										
ClpP	Rbcl	Ycf1	Ycf2	Ycf3	Ycf4										
142	19	161	219	30	39										
ndhA	ndhB	ndhC	ndhD	ndhE	ndhF	ndhG	NdhH	ndhI	ndhJ	ndhK					
339	258	339	293	322	346	335	322	378	340	331					
petA	petB	PetD	petG	petL	petN										
33	15	36	13	71	135										
psaA	psaB	psaC	psaI	psaJ	psaM										
16	10	19	72	24	2214										
psbA	psbB	psbC	psbD	psbE	psbF	psbH	psbI	psbJ	psbK	psbL	psbM	psbN	psbT	psbZ	Psb30
12	18	16	17	21	21	20	18	21	13	22	157	23	22	31	2126
Rpl2	Rpl14	Rpl16	Rpl20	Rpl22	Rpl23	Rpl32	Rpl33	Rpl36							
2	4	3	0	127	24	114	133	5							
Rps2	Rps3	Rps4	Rps7	Rps8	Rps11	Rps12	Rps14	Rps15	Rps16	Rps18	Rps19				
3	3	4	3	3	2	2	7	249	284	5	5

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
