# Peer review of "Gene Loss and Evolution of the Plastome"

_genes, 2020, doi:10.3390/genes11101133_

Round 1

Reviewer 1 Report

Missing some stats or correlations that could lead to more impactful conclusions. You collected all this data which is awesome and needs to be addressed on a large scale but you reported typical information found in plastome papers which is good but leaves something to be desired for me, personally. Is there any correlation between size and CDS regions or between rearrangements in certain groups and larger plastome size or IR-loss, or nucleotide substitutions and rearrangements or IR-loss that you have actually tested using your data? 

Visuals are lacking for me as well – there are a few things that would be easier to see in a table or a figure as opposed to paragraph form.

I feel like some of the supplementary figures are more informative than the figures in the paper

The recombination events figures seem interesting but I’m not sure how to read them so it’s hard to provide feedback.

A few typos throughout - see below. 

Why use OGDraw to determine IR? Was it just easier to visualize or is there a component in OGDraw that allows you to quickly check for the IR?

Line 102 – Why were these genes (PsaM, psb30, ChlB, ChlL, ChlN RPL21) specifically chosen for this analysis?

Line 140 – “The recombination events in 140 IR-deleted and non-deleted IR species were studied separately.” IR-deleted species still have one copy of the IR. Why wouldn’t you just delete one copy of the IR in those species with the second copy and carry out the analyses together? 

Line 191 – typo, “Some other species, however, were contain…”

Line 195 – typo, “…some of the species were encoded lower number of CDS in…”

Scatterplot figs – It would be easier to see the outliers if they were labeled with the species names (only for the extremes, not all of them)

Lines 227-228 – typo, “All of the species those found to possess the PsaM gene were belonged algae, bryophytes, pteridophytes, or…”

Line 289 – confusion, “In the majority of cases, however, the CDS of the Rpl21 genes were truncated. Therefore, only 22 full length CDS were used to identify deletion and duplication events.” What do you mean by this? I don’t see in the methods how you looked for deletions and duplications so I’m confused about how you did that with those 22 full length genes. Did you look for pseudogenes, also?

Line 294 – confusion, “3 RpL21 genes had undergone duplication events, 8 had undergone deletion events, and 9 exhibited co-divergence” Do you mean the RpL21 gene in 3 species had undergone duplication events or something else?

Line 331 – typo, “…were also undergone vivid recombination…”

Section 3.7 – there’s got to be a better way to organize the list of genes as opposed to listing them in a paragraph and then ending with “and others”

Lines 509-515 – “The average size of IR-deleted chloroplast genomes in eudicots, monocots, protists, and gymnosperms was smaller than the average size of chloroplast genomes of taxa where IR regions have not been deleted. Thus, the lower number of CDS in these taxa, may be related to the deletion of IR regions. “

Was the size of the genomes calculated based on including only one copy of the IR in those species with both copies? If not, this may overinflate genome size differences. I would also suggest doing some sort of regression to see the actual trend here.

Author Response

Dear reviewer,

Please find the attached file to get the responses of your comments.

Thanks

Reviewer 2 Report

The authors studied 2511 chloroplast genomes of different species via multiple bioinformatics tools. The analysis revealed that the changes of inverted repeats (IR) sequences in the investigated chloroplast genomes and evolutional characteristics of Rbcl gene and chloroplast-encoded genes across diverse species lineages. Moreover, the authors suggest that chloroplast genomes evolved from multiple common ancestors by analyzing phylogenetic tree of 2511 chloroplast genomes. The results in this work seem to extend our understanding of evolutional aspect of chloroplast genome across different species. However, there are several minor points that should be considered.

Minor comments:

  1. Line 28, “chloroplast encoding genes” needs to be corrected to “chloroplast-encoded genes”
  2. Line 38-42, the authors need to add citations related to the arguments.
  3. Line 49-57, the authors need to add citations related to the arguments.
  4. Line 76, “2357 chloroplast genomes”, “2511 chloroplast genome” is mentioned in the abstract and the result. It should be corrected.
  5. Line 165-167, the number of chloroplast genome in parenthesis is not matched with that of supplemental File 1.
  6. The authors should interpret the evolutional meaning of results from figure 1 to 5 in the maintext.
  7. In the table 1, No. of co-divergence in PsaM gene is not identical to the number written in the line 240. Moreover, Total No. of Sequences Studied is not matched with the total number of duplication, divergence, and losses. The authors need to explain it in detail in the table 1.
  8. Line 326-328, the authors need to add citations related to the arguments.
  9. In the table 2, the gene number of psbN and psbT is not matched with those of supplemental File 4.
  10. Line 395-396, the authors need to add citations related to the arguments.
  11. Line 445-447, the authors need to add citations related to the argument.
  12. In the discussion part, it needs to be improved by adding more references to their arguments.
  13. In the supplementary Data 1, total loss number of the PsbM, ndhF, and ndhJ genes is not matched with the number in parenthesis.
  14. In the supplementary Table 1, “chloroplast encoding genes” needs to be corrected to “chloroplast-encoded genes”

Author Response

(The authors gave the same response as above.)
